# Enhancing Sustainable Mobility: Evaluating New Bicycle and Pedestrian Links to Car-Oriented Industrial Parks with ARAS-G MCDM Approach

Jurgis Zagorskas [1,*] and Zenonas Turskis [2]

1   Department of Engineering Graphics, Faculty of Fundamental Sciences, VILNIUSTECH, Vilnius Gediminas Technical University, Saulėtekio al. 11, 10223 Vilnius, Lithuania
2   The Institute of Sustainable Construction, Faculty of Civil Engineering, VILNIUSTECH, Vilnius Gediminas Technical University, Saulėtekio al. 11, 10223 Vilnius, Lithuania; zenonas.turskis@vilniustech.lt
*   Correspondence: jurgis.zagorskas@vilniustech.lt; Tel.: +370-68814666

**Abstract:** The aim of this research is to address the challenge of transforming car-oriented industrial parks into pedestrian- and bicycle-friendly environments. Through the implementation of a multi-criteria decision-making (MCDM) approach, the study aims to evaluate alternative pathway connections and assess their potential impact on bicycle and pedestrian traffic volumes. By enhancing the connectivity of the cycling pathway network, the research seeks to demonstrate the potential for substantial increases in cycling and walking within industrial zones. This research leverages a multi-criteria decision-making framework, specifically the ARAS-G method, and integrates geographic information system analysis alongside Python scripting to project future bicycle usage and assess alternative pathway connections. The study underscores the potential for substantial increases in cycling and walking by augmenting the connectivity of the cycling pathway network. The findings hold practical significance for urban planners and industrial zone developers, advocating a holistic approach to sustainable transportation. The research contributes a comprehensive set of criteria encompassing connectivity, safety, accessibility, efficiency, integration within the urban fabric, and cost-effectiveness to evaluate sustainability and prioritize actions and measures for reestablishing industrial zones as bicycle-friendly spaces.

**Keywords:** ARAS-G; GIS analysis; cycling; MCDM; micro-mobility; sustainable transportation

## 1. Introduction

The urgent need to address global greenhouse gas emissions, which are significantly attributed to motorized transportation (accounting for up to 16%), has prompted countries worldwide to commit to achieving carbon neutrality and decarbonizing the transport sector [1]. In line with these sustainability goals, the European Union has set an ambitious target of reducing greenhouse gas emissions by 90% by 2050, and countries such as the UK aim to have 50% of trips made by walking and cycling in towns and cities by 2030 [2]. Achieving these objectives necessitates a shift towards sustainable active modes of travel, such as walking, wheeling, and cycling [3].

As transportation planning progresses from strategy to implementation, localized planning interventions have gained prominence in both policy and academic discourse [4]. Researchers and policymakers in the transportation field are actively seeking practical solutions to achieve carbon neutrality through modal shifts [5,6]. Cities across Europe are making substantial investments in cycling and walking infrastructure to promote sustainable travel. Notably, countries like the Netherlands, Denmark, Sweden, and other regions where cycling is a prevalent mode of transportation have achieved remarkable success.

The introduction of electric-powered personal mobility vehicles (e-PMVs) has further transformed travel patterns [7,8]. Shared e-scooters, in particular, have significantly

impacted cycling path traffic volume, accounting for nearly 50% of overall cycling traffic since 2019 [8]. It is important to check the sources of electricity. While existing research primarily relies on shared e-scooter trip data [9–11], it is worth noting that personally used e-PMVs enable faster speeds (30–35 km/h) and longer travel distances (7 km) compared to traditional human-powered bicycles, expanding the possibilities for micro-mobility options [11].

The COVID-19 pandemic and associated restrictions have also amplified interest in cycling and walking [12]. Governments have responded by reallocating streets and transforming vehicle lanes and parking spaces into temporary walking or cycling paths [13]. Should these changes become permanent, they hold substantial implications for population health, travel behavior, and transport equity [14].

While cycling promotion and infrastructure development are prioritized in urban centers, determining where and when to invest in cycling infrastructure remains a challenge. Understanding travel demand between residential areas and job sites, especially concerning industrial zones, is critical, yet studies have not precisely estimated these connections. Many European cities require more links between industrial zones and residential areas due to planning decisions that have historically separated these zones from living spaces. However, as heavy-polluting industries have relocated to other regions, these zones now have the opportunity to establish these missing links. Rebuilding and reorganizing extensive industrial territories present an opportunity to consider transport decarbonization targets and offer accessibility beyond traditional automobile modes.

The freight and passenger transport sector is poised to make significant strides in reducing greenhouse gas (GHG) emissions. Electromobility is emerging as a viable alternative to traditional mobility systems throughout the EU [15]. Assessing GHG emissions from electric vehicles and comparing GHG savings relative to internal combustion engine vehicles hinge on understanding the carbon intensity of the electricity used to recharge electric vehicle batteries [16].

Despite the growing prevalence of electric-drive vehicles, internal combustion engines remain predominant as power units in mobile systems across diverse economic sectors, especially in industry. However, reducing emissions, particularly NOx and PM, is imperative due to their adverse impact on human health and the environment's pollution with harmful exhaust gases [17]. To mitigate exhaust gas emissions and comply with legal regulations, hybrid drives featuring optimized internal combustion engines and associated systems are increasingly deployed as interim solutions. This underscores the ongoing importance of addressing emissions from internal combustion engines, particularly in industrial zones where pedestrian and cyclist safety is paramount. Diesel engines hold potential as power units for hybrid vehicles due to their good fuel economy [18].

While progress has been made in constructing missing links in many cities, there is a need for more comparative studies on alternative bicycle pathway segments, priority lists, or estimations of changes in travel time and costs following the establishment of new links. This situation can be attributed to traditional transport planning methods that calculate traffic flows between transportation districts without considering local street segment-level changes, but traditional methods are not sensible to micro-level or pathway segment-level changes. The present generation of traffic planners often relies on macro-level data, despite advancements in processing speed and data capacity [19–22]. Micro-simulation, typically performed with software such as PTV VISIM, analyses changes in existing traffic flows, link delays, volumes, and traffic light phases [23,24], but often fails to provide detailed predictions at the neighborhood level.

This study aims to address these gaps by introducing a pioneering approach that encompasses various analytical objectives. It not only facilitates the computation of connectivity using prediction of traffic volumes for non-existing links to support investment strategies but also employs the novel Multi-Criteria Decision-Making (MCDM) method, ARAS-G, characterized by its distinctive feature of incorporating Grey theory. ARAS-G

represents an innovative tool that effectively grapples with the inherent uncertainties of decision-making processes, particularly in the context of sustainable urban mobility.

Grey theory is a robust method revered for its competence in addressing uncertainty, notably excelling in the mathematical analysis of systems with incomplete or uncertain data. Its preeminence over fuzzy sets theory lies in its adeptness at accommodating situations riddled with vagueness and indeterminacy.

Nevertheless, information collected from real-world scenarios invariably presents itself as uncertain and frequently incomplete. Consequently, the expansion of application domains from traditional white numbers, characterized by crisp values, to grey numbers becomes a requisite undertaking for authentic real-world scenarios. ARAS-G emerges as a quintessential fusion of MCDM and Grey theory, giving decision–makers the capacity to navigate complex and ambiguous situations. This innovative integration contributes to the advancement of rigorous and nuanced decision-making strategies in the face of pervasive uncertainties and information gaps, thereby bolstering the prospects of sustainable urban mobility planning.

This study seeks to address these gaps by presenting a practical method for various analysis purposes, enabling connectivity calculations in real traffic volume numbers and predicting traffic volumes for non-existing links to support investment strategies. This method utilizes standard and affordable GIS software (QGIS 3.34.4 'Prizren' for Windows and Python libraries for network graph analysis), eliminating the need for specialized expertise or computer programming skills. It leverages an origin–destination matrix generated from GIS point layers and a network graph derived from OpenStreetMap (OSM) GIS data.

## 2. Aims and Scope of Research

The primary objectives of this research include the following:

- Addressing the Challenge with the ARAS-G MCDM for alternative selection: This study aims to bridge gaps by introducing a practical approach for various analytical purposes, primarily focusing on using the innovative Multi-Criteria Decision-Making (MCDM) method ARAS-G. The research employs ARAS-G for the selection of the most suitable alternatives in the context of enhancing pedestrian and bicycle connectivity in car-oriented industrial parks. This method is adeptly applied to facilitate the systematic and objective evaluation of alternative solutions; it aids in selecting the most appropriate interventions for transforming car-oriented industrial parks into pedestrian- and bicycle-friendly environments;
- Analyzing and Predicting Bicycle Flows through Micro-Scale Analysis: This research undertakes an in-depth analysis of bicycle flows and forecasts, primarily relying on micro-scale origin–destination (OD) analysis;
- Evaluating pathway connections: In the process, the study evaluates the efficacy of alternative pathway connections to determine their potential impact on enhancing bicycle and pedestrian connectivity. This entails assessing the performance of different pathway connections while considering key criteria;
- Demonstrating the potential for increases in cycling and walking: The ultimate aim is to highlight the potential for significant increases in cycling and walking within car-oriented industrial parks by improving pathway connectivity. Through the selection of alternatives facilitated by ARAS-G and comprehensive micro-scale analysis, the research aims to substantiate the viability of these improvements.

This research is poised to deliver valuable insights and contribute to the discourse on sustainable urban mobility. By utilizing ARAS-G for alternative selection and focusing on micro-scale OD analysis for forecasting and evaluation, the study endeavors to provide a rigorous and data-driven approach to transforming car-oriented industrial parks into pedestrian- and bicycle-friendly environments. The research outcomes contribute to society's well-being, environmental sustainability, and the creation of more livable cities by advocating for the integration of pedestrian- and bicycle-friendly environments.

## 3. Recent Research and Literature Review

Micro-mobility, which encompasses lightweight shared transportation solutions within urban environments, has garnered significant attention as a means to address issues such as traffic congestion, environmental considerations, and last-mile transportation challenges. The integration of multi-criteria decision-making (MCDM) approaches into modeling and simulation techniques is crucial for comprehensively understanding the intricate dynamics of micro-mobility systems and for formulating effective implementation strategies. This literature review aims to provide an extensive overview of recent developments in micro-mobility modeling and simulation research, with a specific emphasis on the utilization of MCDM techniques. It also highlights the main methodologies, challenges, and future research directions in this field.

In the realm of urban planning, decisions regarding planned bicycle and pedestrian infrastructure are frequently made without rigorous calculations. This practice is rooted in various financial and professional considerations, with one of the key factors being the absence of supportive methodologies. As many cities grapple with the inadequacy of cycling infrastructure networks, policymakers are confronted with the task of designing interconnected bicycle facility networks to ensure safe and convenient access for cyclists to various urban destinations. This not only promotes infrastructure utilization but also enhances the overall usability of these facilities [25,26].

While there is ample evidence illustrating the multifaceted impacts of infrastructure on demand, only a limited number of studies have delved into the combined cost-benefit performance of bicycle infrastructure. A recent study from Denmark, for instance, revealed that the benefits derived from investments in bicycle infrastructure translate into an impressive internal rate of return of 11% annually [27]. This figure significantly surpasses the returns associated with most other public transportation and car infrastructure investments.

Conversely, industrial zones are typically strategically located with optimal access to road, rail, or maritime transportation. In European countries, these zones are often situated on the outskirts of core cities to minimize pollution in residential areas. However, contemporary industry zones within the European Union tend to produce minimal pollution, with the majority of environmental impacts stemming from heavy freight transportation. Regrettably, many industrial zones are poorly connected to cycling and pedestrian pathways due to their considerable distance from residential areas. This issue is seldom addressed in traffic planning, despite a substantial portion of daily commuting transpiring between residential districts and industrial zones. While studies have examined noise and air pollution stemming from traffic or traffic congestion in industrial zones [28,29], they rarely explore non-pollutant modes of transportation, such as cycling, wheeling, and walking.

Several proposals have surfaced concerning eco-friendly solutions and the revitalization of industry zones within their boundaries [30,31]. Nevertheless, few studies have been dedicated to reconnecting industrial zones with urban cores.

One recent concept gaining traction is the "15-minute" or "20-minute city," advocated by New Urbanism and Smart Growth approaches [32,33]. At its core, this concept revolves around the idea that most essential needs can be satisfied within a 15-minute round trip from one's residence, often equating to a 10-minute walk or cycling and public transport in certain definitions [32,34,35]. The distance between residential and working areas takes center stage in this concept. Achieving a 15-minute commute from residential zones to industrial areas poses a significant challenge; other amenities in European cities are often already optimally located [34].

### Micro-Mobility Modeling and Simulation

Multiple micro-mobility modeling approaches can be categorized into three main groups, namely Agent-Based Modeling (ABM), Network-based Modelling, and Data-Driven Modelling. Authors may combine these groups for various purposes, and the integration of Multi-Criteria Decision-Method (MCDM) approaches can further enhance the analysis and decision-making process in micro-mobility modeling [36].

Agent-based modeling allows for the representation of individual actors and their interactions within a micro-mobility system. Recent studies have highlighted the integration of MCDM approaches within ABM frameworks to account for multiple criteria, such as accessibility, equity, and environmental impact, when evaluating the performance of micro-mobility systems [37].

Network-based models represent micro-mobility systems as a network of nodes and links, considering factors such as origin–destination flows, route choice behavior, and network topology. The application of MCDM techniques in network-based modeling can support decision-making related to infrastructure planning, resource allocation, and system optimization in micro-mobility contexts [38–44].

Data-driven modeling techniques leverage abundant data from micro-mobility service providers, urban sensors, and social media. Recent research highlights the integration of MCDM methods with data-driven models to consider multiple criteria, such as demand patterns, service coverage, and user satisfaction, for effective decision making in micro-mobility planning and management [38,45–47].

In the field of transportation modeling, the conventional practice involves employing a mezzo-scale approach wherein territories are partitioned into larger transport zones for the sake of simplifying calculations. The primary software utilized for this purpose is PTV VISUM or VISIM. However, these commercial software packages primarily focus on tasks related to traffic regulation, traffic signal adjustment, and automobile-centric mobility.

Remarkably, micro-mobility modeling and prediction in commercial software have received limited attention thus far, primarily due to the considerable increase in the scope and complexity of calculations involved. It is only in recent years that emerging approaches towards predicting individual mobility have surfaced, thanks to advancements in individual data collection methods and computing techniques. The advancements in computational power and the availability of cloud services for interactive computing have also greatly facilitated the evaluation of prediction models on a large scale and with high data demand.

Over the last decade, traffic in large urban areas and road networks have been simulated also using ABMs. These models usually simulate the impacts and the overall system performance of innovative transport modes, such as shared autonomous vehicles, electric taxis, demand-responsive buses, electric scooters, etc.. For micro-simulations, "TRANSIMS", "MATSim", "NetLogo", "SimMobility", "Anylogic", and "SARL" are considered the most used [48]. They give very detailed simulations of travel volumes in time and space. These methods can be used for predicting travel demand, but mainly they are used to predict the traffic in peak hours, manage traffic congestion, and synchronize the traffic lights.

Recent studies provide methods to forecast cycling/pedestrian flow from mobile devices, STRAVA and other web platforms [49–52], but these methods are limited to work only on existing infrastructure.

There are numerous research on using Multi-Criteria Decision Methods (MCDM) for finding the best locations for mobility hubs [40,42,53], shared bicycle locations [54,55], path segment construction priorities, etc.

Existing methods for evaluating network connectivity and choice on the micro-scale are mostly Space Syntax (SS)-based tools to define the central nodes and segments in the path network [56–58]. However, SS methodology does not take into account the people connected to the network. Therefore, the obtained analysis results have weak supporting arguments behind them, although mathematically they are correct. More geographically localized data on each building with numbers of living places, working places or number of visitors, have to be added to SS to provide a sufficient basis for urban development scenarios.

## 4. Materials and Methods

### 4.1. Establishing a Criteria Framework for the Multi-Criteria Decision-Making (MCDM) Approach

In order to develop a comprehensive system for evaluating connection alternatives, an expert group consisting of transportation and urban planning specialists from Vilniustech,

in collaboration with town planners, were briefed on available research data and methods for predicting traffic volumes. Through a series of discussions, a set of 10 criteria was meticulously devised, ensuring that the criteria values could be accurately estimated through proposed calculations.

The criteria system was meticulously delineated, and the weights of each criterion were once again determined by experts, as illustrated in Table 1. The selected criteria include:

**Table 1.** Expert rankings for criteria.

|  | $E_1$ | $E_2$ | $E_3$ | $E_4$ | $E_5$ | $E_6$ | $E_7$ | $E_8$ | $E_9$ | $E_{10}$ | $\Sigma$ | w |
|---|---|---|---|---|---|---|---|---|---|---|---|---|
| $x_1$ | 9 | 9 | 9 | 9 | 8 | 9 | 9 | 9 | 9 | 9 | 89 | 0.14 |
| $x_2$ | 8 | 8 | 8 | 8 | 8 | 9 | 8 | 8 | 8 | 8 | 81 | 0.13 |
| $x_3$ | 7 | 7 | 8 | 7 | 7 | 8 | 8 | 7 | 7 | 8 | 74 | 0.12 |
| $x_4$ | 7 | 7 | 8 | 7 | 7 | 8 | 8 | 7 | 7 | 8 | 74 | 0.12 |
| $x_5$ | 6 | 7 | 6 | 6 | 6 | 6 | 6 | 7 | 6 | 6 | 62 | 0.10 |
| $x_6$ | 5 | 5 | 6 | 5 | 5 | 6 | 6 | 5 | 5 | 6 | 54 | 0.08 |
| $x_7$ | 5 | 5 | 5 | 5 | 5 | 5 | 5 | 5 | 5 | 5 | 50 | 0.08 |
| $x_8$ | 4 | 4 | 5 | 4 | 4 | 5 | 5 | 4 | 4 | 5 | 44 | 0.07 |
| $x_9$ | 3 | 4 | 3 | 4 | 4 | 4 | 4 | 3 | 3 | 4 | 36 | 0.06 |
| $x_{10}$ | 3 | 3 | 3 | 3 | 3 | 3 | 3 | 3 | 3 | 3 | 30 | 0.05 |
|  |  |  |  |  |  |  |  |  |  | $\Sigma$ | 642 | 1 |

$X_1$—Population within the attractive distance to use the bicycle or electricity-powered personal mobility vehicles. The distance considered was 5 to 7 km; therefore, this criterion has two numbers or a G value number.

$X_2$—Average travel distance from all residential areas to workplaces within a 7 km radius, expressed in meters.

$X_3$—Estimated number of daily non-motorized commuters, denoted in terms of the percentage of total individuals within the 7 km catchment area, with scenarios accounting for both 5 percent and 15 percent usage rates.

$X_4$—Reduction in traffic flow on the A6 road, quantified in terms of the number of vehicles, resulting from the increased adoption of non-motorized commuting methods.

$X_5$—Financial savings derived from reduced greenhouse gas emissions, measured in million euros per year, attributable to the greater utilization of bicycles or electric personal mobility vehicles.

$X_6$—Total distance traveled via non-motorized means, expressed as 1000 km per year per passenger, reflecting the increased accessibility of workplaces from residential areas.

$X_7$—Financial savings arising from reduced traffic congestion, measured in million euros per year, stemming from alleviated congestion along the A6 route due to heightened bicycle commuting.

$X_8$—Financial savings resulting from reduced travel costs, measured in million euros per year, consequent to the preference for bicycles over other modes of transportation for commuting purposes.

$X_9$—Construction costs associated with alternative connections, quantified in million euros. These cost estimates were sourced from the 2022 annual traffic construction bulletin from Poland, where similar constructions have been undertaken. The data indicate that the average cost for constructing either a viaduct or tunnel crossing a 50 m wide highway is approximately 1.25 million euros. Additionally, the construction costs for additional connecting bicycle lanes were estimated at 50,000 euros per kilometer and vary for each alternative. When evaluating projects, a more thorough analysis of construction costs must be conducted. $X_{10}$—Risk of accidents at hazardous A6 crossings, determined by the

projected number of potential non-vehicle commuters on this route, serving as a proxy for the perceived level of danger. Lower values indicate lower perceived risk levels, with increased commuter presence correlating with heightened accident risk.

These criteria collectively serve as the foundation for evaluating the efficacy and impact of interventions or policies aimed at promoting bicycle commuting, with considerations spanning economic viability, environmental sustainability, congestion mitigation, and the safety of individuals traversing the A6 route.

Experts ranked the criterion $X_1$—Population within the attractive distance as the most important for this type of research. Other important criteria were $X_2$—average travel distance, $X_3$—number of daily non-motorized commuters, and $X_4$—Reduction of traffic flow. While financial savings, safety, and construction costs were deemed less pivotal, they remained important considerations. Expert consensus suggested that the construction costs of such measures typically recoup within 5 to 10 years, mitigating the necessity of prioritizing this criterion as extensively as in traditional cost-effectiveness analyses.

*4.2. Grey Numbers*

Grey theory serves as a methodology for scrutinizing uncertainty, demonstrating particular prowess in mathematically analyzing systems imbued with uncertain data. Notably, grey theory surpasses fuzzy sets theory by adeptly managing situations marked by fuzziness. The process of alternative selection mirrors a grey system process, amenable to effective resolution through grey theory methodologies. Criteria assessments are delineated using linguistic variables, feasibly represented through grey numbers. Grey theory, first introduced by Deng [59], offers a robust mathematical framework for:

- Addressing issues characterized by incomplete information;
- Overcoming the inherent limitations of conventional statistical methods;
- Utilizing a limited dataset to predict the behavior of uncertain systems, especially when dealing with discrete data and incomplete information [60].

White numbers, grey numbers, and black numbers are three classifications used to discern the degree of uncertainty in information [61]. Let

$$\otimes x = [\alpha, \ \gamma] = \{x | \alpha \leq x \leq \gamma, \ \alpha \ \text{and} \ x \in R\}. \tag{1}$$

Next, a grey number, denoted as $\otimes x$ with two real numbers $\alpha$ (the lower limit) and $\gamma$ (the upper limit) is defined as follows:

- If $\alpha \to -\infty$ and $\gamma \to \infty$, then $\otimes x$ is called a black number, indicating a lack of meaningful information;
- Else if $\alpha = \gamma$, then $\otimes x$ is referred to as a white number, indicating complete information;
- Otherwise, $\otimes x = [\alpha, \ \gamma]$ is termed a grey number, indicating insufficient and uncertain information.

However, information derived from real-world sources inherently carries uncertainty or incompleteness. Thus, broadening the scope from precise white numbers (crisp values) to grey numbers becomes imperative for real-world applications. The fundamental definitions and operations of grey numbers are outlined as follows.

Let a grey number be defined as a grey number defined by two parameters $(\alpha, \ \gamma)$. Let $+, \ -, \ \times$ and $\div$ denote the operations of addition, subtraction, multiplication and division, respectively. The basic operations of grey numbers $\otimes n_1$ and $\otimes n_2$ are defined as follows:

$$\otimes n_1 + \otimes n_2 = \left(n_{1\alpha} + n_{2\alpha}, n_{1\gamma} + n_{2\gamma}\right) \ addition \tag{2}$$

$$\otimes n_1 - \otimes n_2 = \left(n_{1\alpha} - n_{2\gamma}, n_{1\gamma} - n_{2\alpha}\right) \ subtraction \tag{3}$$

$$\otimes n_1 \times \otimes n_2 = \left(n_{1\alpha} \times n_{2\alpha}, \ n_{1\gamma} \times n_{2\gamma}\right) \ multiplication \tag{4}$$

$$\otimes n_1 \div \otimes n_2 = \left(\frac{n_{1\alpha}}{n_{2\gamma}}, \frac{n_{1\gamma}}{n_{2\alpha}}\right) \ division \tag{5}$$

$$k \times (\otimes n_1) = \left(kn_{1\alpha} \,,\, kn_{1\gamma}\right) \text{ Number product of grey numbers if } k \text{ is positive real number} \quad (6)$$

$$(\otimes n_1)^{-1} = \left(\frac{1}{n_{1\gamma}} \,,\, \frac{1}{n_{1\alpha}}\right) \quad (7)$$

*4.3. The Proposed Grey MCDM: An Additive Ratio Assessment Method with Grey Values (ARAS-G)*

The ARAS method, as outlined in [62], is rooted in the premise that understanding complex phenomena in our world can be achieved through straightforward relative comparisons. It posits that the degree of optimality attained by a compared alternative is determined by the ratio of the sum of normalized and weighted values of the criteria describing the alternative in question to the sum of the values of normalized and weighted criteria characterizing the optimal alternative.

In accordance with the ARAS method, the value of the utility function, which assesses the complex relative efficiency of a feasible alternative, correlates directly with the relative influence of the values and weights assigned to the key criteria evaluated in a given project. The initial step involves forming a Grey Decision-Making Matrix (GDMM). In the context of the Generalized Multi-Criteria Decision-Making (GMCDM) framework for discrete optimization problems, any problem at hand is represented by a GDMM that accounts for preferences across m reasonable alternatives (rows) evaluated against n criteria (columns):

$$\widetilde{X} = \begin{bmatrix} \otimes x_{01} & \cdots & \otimes x_{0j} & \cdots & \otimes x_{0n} \\ \vdots & \ddots & \vdots & \ddots & \vdots \\ \otimes x_{i1} & \cdots & \otimes x_{ij} & \cdots & \otimes x_{in} \\ \vdots & \ddots & \vdots & \ddots & \vdots \\ \otimes x_{m1} & \cdots & \otimes x_{mj} & \cdots & \otimes x_{mn} \end{bmatrix}; \quad (8)$$
$$i = \overline{0,\, m}; \quad j = \overline{1,\, n},$$

where *m*—number of alternatives, *n*—number of criteria describing each alternative, $\otimes x_{ij}$—grey value representing the performance value of the *i* alternative in terms of the *j* criterion, $\otimes x_{0j}$—optimal value of *j* criterion.

If the optimal value of the *j* criterion is unknown, then

$$\begin{aligned} \otimes x_{0j} &= \max_i \otimes x_{ij}, & \text{if } \max_i \otimes x_{ij} \text{ is preferable, and} \\ \otimes x_{0j} &= \min_i \otimes x_{ij}^*, & \text{if } \min_i \otimes x_{ij}^* \text{ is preferable.} \end{aligned} \quad (9)$$

Typically, the performance values $\otimes x_{ij}$ and the criteria weights $\otimes w_j$ are represented as entries within a Decision-Making Matrix (DMM). Experts determine the criteria system, values, and initial weights, with potential adjustments based on stakeholders' objectives and opportunities. The prioritization of alternatives unfolds across multiple stages, especially considering that criteria often vary in dimensions.

Subsequently, the focus shifts to deriving dimensionless weighted values from comparative criteria, aiming to mitigate challenges stemming from differing criteria dimensions. To achieve this, the ratio to the optimal value is commonly employed. While various theories exist on this ratio, the values are typically mapped onto either a specified interval [0; 1] or [0; ∞] through the normalization process of the DMM. In the subsequent stage, all criteria's initial values undergo normalization, defining the values of the normalized

decision-making matrix. The initial values of all the criteria are normalized—defining values $\otimes\overline{x}_{ij}$ of the normalized decision-making matrix $\otimes\overline{X}$:

$$
\otimes\overline{X} =
\begin{bmatrix}
\otimes\overline{x}_{01} & \cdots & \otimes\overline{x}_{0j} & \cdots & \otimes\overline{x}_{0n} \\
\vdots & \ddots & \vdots & \ddots & \vdots \\
\otimes\overline{x}_{i1} & \cdots & \otimes\overline{x}_{ij} & \cdots & \otimes\overline{x}_{in} \\
\vdots & \ddots & \vdots & \ddots & \vdots \\
\otimes\overline{x}_{m1} & \cdots & \otimes\overline{x}_{mj} & \cdots & \otimes\overline{x}_{mn}
\end{bmatrix}; \qquad (10)
$$

$$
i = \overline{0,\ m};\ \ j = \overline{1,\ n}.
$$

The criteria, whose preferable values are maxima, are normalized as follows:

$$
\otimes\overline{x}_{ij} = \frac{\oplus x_{ij}}{\sum\limits_{i=0}^{m} \otimes x_{ij}}. \qquad (11)
$$

The criteria, whose preferable values are minima, are normalized by applying a two-stage procedure:

$$
\otimes x_{ij} = \frac{1}{\otimes x_{ij}^{*}}; \quad \otimes\overline{x}_{ij} = \frac{\otimes x_{ij}}{\sum\limits_{i=0}^{m} \otimes x_{ij}}. \qquad (12)
$$

When the dimensionless values of the criteria are known, all the criteria, originally having different dimensions, can be compared.

The next stage is defining the normalized-weighted matrix—$\otimes\hat{X}$. It is possible to evaluate the criteria with weights $0 < \otimes w_j < 1$. Only thoroughly grounded weights can be utilized, as they inherently carry subjectivity and significantly impact the resultant solution. The values of weight $\otimes w_j$ are usually determined by the expert evaluation method. The sum of weights $w_j$ would be limited as follows:

$$
\sum_{j=1}^{n} w_j = 1. \qquad (13)
$$

$$
\otimes\hat{X} =
\begin{bmatrix}
\otimes\hat{x}_{01} & \cdots & \otimes\hat{x}_{0j} & \cdots & \otimes\hat{x}_{0n} \\
\vdots & \ddots & \vdots & \ddots & \vdots \\
\otimes\hat{x}_{i1} & \cdots & \otimes\hat{x}_{ij} & \cdots & \otimes\hat{x}_{in} \\
\vdots & \ddots & \vdots & \ddots & \vdots \\
\otimes\hat{x}_{m1} & \cdots & \otimes\hat{x}_{mj} & \cdots & \otimes\hat{x}_{mn}
\end{bmatrix}; \qquad (14)
$$

$$
i = \overline{0,\ m};\ \ j = \overline{1,\ n}.
$$

Normalized-weighted values of all the criteria are calculated as follows:

$$
\otimes\hat{x}_{ij} = \otimes\overline{x}_{ij} \times \otimes w_j; \qquad\qquad i = \overline{0,\ m}, \qquad (15)
$$

where $w_j$ is the weight (importance) of the $j$ criterion and $\overline{x}_{ij}$ is the normalized rating of the $j$ criterion.

The following task is determining values of the optimality function:

$$
\otimes S_i = \sum_{j=1}^{n} \otimes\hat{x}_{ij}; \qquad\qquad i = \overline{0,\ m}, \qquad (16)
$$

where $\otimes S_i$ is the value of the optimality function of the $i$ alternative.

The highest value represents the optimal outcome, while the lowest signifies the least favorable. Considering the computational procedure, the optimality function $\otimes S_i$ has a

direct and proportional relationship with the values $\otimes x_{ij}$ and weights $\otimes w_j$ of the examined criteria and their respective impact on the ultimate outcome. Hence, the higher the value of the optimality function $\otimes S_i$, the more effective the alternative. The prioritization of alternatives can be established based on the value $\otimes S_i$. Therefore, this method proves advantageous for assessing and ranking decision alternatives.

The result of grey decision making for each alternative is the grey number $\otimes S_i$. Various techniques exist for converting grey values into crisp values. Among them, the center-of-area method stands out as the most practical and straightforward to implement:

$$S_i = \frac{1}{2}(S_{i\alpha} + S_{i\gamma}). \tag{17}$$

The utility of the alternative is assessed by comparing it to the ideally best one $S_0$.

The equation used for the calculation of the utility degree $K_i$ of an alternative $A_i$ is given below:

$$K_i = \frac{S_i}{S_0}; \qquad i = \overline{0, \; m}, \tag{18}$$

where $S_i$ and $S_0$ are the optimality criterion values, obtained from Equation (16).

It is evident that the calculated values $K_i$ fall within the interval [0; 1] and can be arranged in an increasing sequence, as desired.

*4.4. The Micro-Scale Origin–Destination Method for Forecasting Bicycle Flows on Links That Do Not Currently Exist*

This case study presents a methodology for forecasting micro-scale bicycle and pedestrian flows, utilizing easily collectable data and affordable software. This method can be applied to diverse urban contexts. The network is represented as a bidirectional Graph based on existing and proposed bicycle path segments within the area. To assess the usability of network segments, an Origin–Destination (OD) matrix is constructed using the nodes of the Network Graph. For each graph node, the cumulative counts of the nearest employment and residential destinations are computed. Subsequently, the gravity method is employed, incorporating various threshold travel distances to fine-tune the decay function, as outlined in Equation (19) below.

$$\mathbf{O}_n = \sum\nolimits_{j \in J, i \in I} S_j D_i f(c_{ij}) \tag{19}$$

where

$\mathbf{O}_n$ is the cumulative opportunity of use of network segment $n \in N$,
$S_j$ is the weight of supply facility (housing) at node $j \in J$,
$D_i$ is the weight of demand facility (jobs) at node $i \in I$,
$f(\ldots)$ is the distance decay function,
$c_{ij}$ is the travel distance between locations $i$ and $j$.

To calculate the real number of bicycle commuters, Formula (20) is used, where the town population, accessible population employment-to-population ratio, travel mode ratio, and total number of jobs in the area are used for calculation.

$$\mathbf{N}_c = \frac{P_a}{P} \cdot E \cdot k_C \cdot D \tag{20}$$

where

$\mathbf{O}_n$ is the number of commuters passing daily through the network segment $n \in N$,
$P$ is the population in the town or adjacent territories from where the workers will arrive,
$E$ is the employment-to-population ratio,
$P_a$ is the number of people who can access the job places at a fixed distance calculated by Equation (20),
$k_c$ is the travel mode coefficient for the cycling travel mode (desired or measured),

*D* is the total number of jobs in the destination area.

The number of people living in proximity to jobs is calculated similarly to $O_n$ in Equation (19), but it is normalized by the total number of jobs and a number $P_a$ is derived, characterizing all the network N; see Equation (21). $P_a$ is calculated as an integral equation, because the jobs are distributed in the territory and situations occur where only part of jobs can be reached in a specified distance.

$$P_a = \sum_{j \in J, i \in I} S_j D_i f(c_{ij}) / D \tag{21}$$

where

$P_a$ is the number of people who can access the job places at a fixed distance. $P_a$ shows the number of people who can access the jobs fully or partially; it is normalized to 1 unit, given if full coverage of jobs is achieved by a specified distance.

$S_j$ is the weight of supply facility (housing) at node $j \in J$,

$D_i$ is the weight of demand facility (jobs) at node $i \in I$,

$f(\ldots)$ is the distance decay function,

$c_{ij}$ is the travel distance between locations *i* and *j*.

*D* is the total number of jobs in the destination area.

*4.5. Data Sources and Software for the Modeling*

The data used for this study were of two types: GIS vector layers of buildings and bicycle pathways and streets, and official population registry as well as legal entities and employees registry to obtain locations and quantities of living and working places.

Subsequently, the data were integrated into the Geographic Information System (GIS), wherein the locations were linked to a graph generated from the bicycle pathway network.

The software used was freeware QGIS 3.34.4 'Prizren' for Windows with Python libraries for graph analysis and mathematical operations Networkx, Scipy, and Numpy.

## 5. The Facts about Kaunas Free Economic Zone (FEZ)

Kaunas Free Economic Zone (FEZ) is located near the A6 and A1 motorways in close proximity to Kaunas town, Lithuania. It is a 534-hectare industrial development area which offers favorable taxes for production and logistics companies. One third of the territory has already developed transportation and infrastructure. The territory of Kaunas FEZ is huge; it is only starting to build up and has potential for further development. It has a perfect position for freight logistics and connection to skilled workforce coming from Kaunas city.

Kaunas FEZ was established in 1996, but has been used for this purpose only from 2005. Kaunas FEZ has attracted 680 million euros of direct investment from its opening in 2005. Businesses operating in Kaunas FEZ territory have some tax relief. Companies are exempt from real estate tax until 2045, do not pay income tax for 10 years, and pay half of the income tax for the next 6 years. Only big companies that invest at least 100 thousand euros and have at least 20 employees are allowed to settle here. By the year 2023, 38 foreign and Lithuanian companies were operating in Kaunas FEZ. There are around 5500 working places created in the territory.

The territory is generally accessible by car and there is only one public bus route passing beside the territory. Bicycle infrastructure inside the territory is persistent, newly built, but there are no proper connections to Kaunas town.

Although the territory is in a strategic location, during the morning and evening rush hours the traffic congestion around the A1–A6 road crossing is the worst in the whole Kaunas region. Especially during the evening rush hours (17:00–18:30) the main roads are blocked for about 1–2 h daily. It is caused mainly by private cars and heavy freight transport. In recent years the traffic problems have been increasing. Alternative travel modes for the FEZ workers coming from Kaunas town would be an essential mean to reduce the congestion. It would also be a popular solution amongst workers, who are now spending a few hours daily in traffic jams.

As Figure 1 shows, the existing bicycle path runs along the A6 road, crossing the motorway. Then the cyclist has to cross the A6 by viaduct to enter the FEZ territory only in the middle, after already passing half of its functioning territory. This connection is unsafe for cyclists.

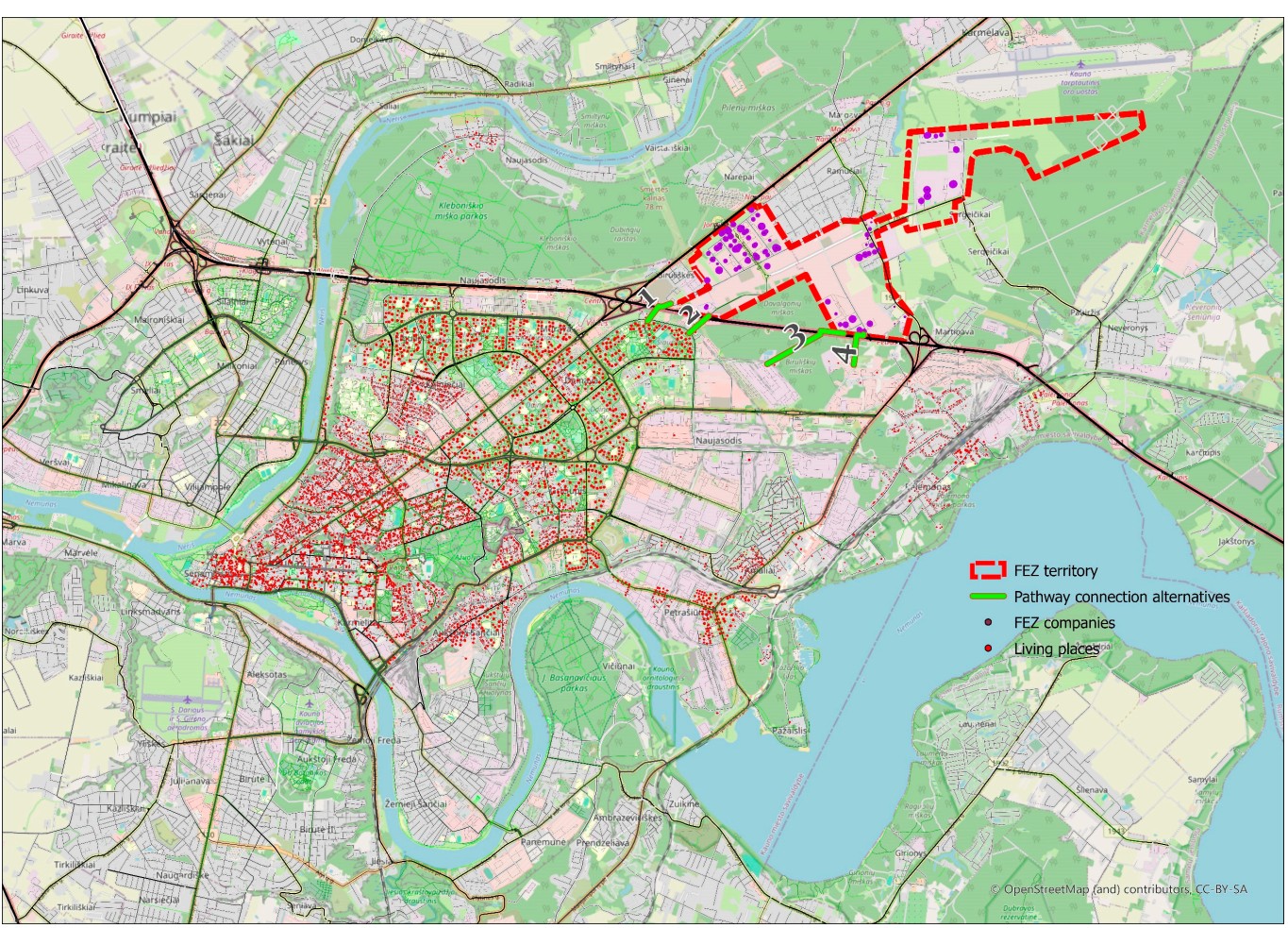

**Figure 1.** Bicycle connections to Kaunas FEZ territory (red dashed line marks the FEZ territory, green bold lines with numbers—proposed highway crossing connections 1 to 4 as viaducts or tunnels).

## 6. Calculations and Results

The Graph of the bicycle pathway network for the case study of Kaunas FEZ and town territory analysis was created from combined OSM layers for roads, cycle paths and pedestrian walkways. The geometry was simplified to create a network Graph consisting of 538 nodes and 610 edges with real distance attributes attached to each edge. This attribute is used for shortest distance calculations later.

The exact data of living places and jobs were collected from the registry. Only the upper part of the Kaunas town living area was selected, because other parts are problematic to reach from Kaunas FEZ. This area contains 8896 point entities representing housing units with an attribute value for the number of people living in the building. The total number of people living in this area is 210,958 and corresponds to 71% of the Kaunas town population (297,214). The other 29% of inhabitants live in areas disconnected from Kaunas FEZ by physical barriers and distance; therefore, they could not be considered as cycling commuters.

Jobs data are taken from the registry and represented by 97 point entities in the FEZ territory with an attribute of the number of jobs in each building location. The total number of jobs is 5642.

To simplify calculations, the numbers from population point entities as well as jobs point entities are summed up to the nearest Graph nodes and an Origin–Destination matrix with weight values is created. Not all the nodes receive housing or jobs numbers; some of the Graph nodes serve only as connecting nodes. There are 388 nodes with non-zero housing (origin) weights and only 50 nodes with non-zero weights for jobs (destination). Figure 1 represents the scheme of the created Graph with Origin—Destination nodes marked by proportional size symbols.

To examine the existing situation and evaluate the improvements made by adding safe crossing tunnels or viaducts across the A1 motorway, the status quo situation and expanded bicycle pathway network situations are examined with distances of 5000 m and 7000 m.

Figure 2 demonstrates the traffic flow graph in the status quo situation with 5 km and 7 km estimation distances. The schemes show overused unsafe bicycle path segments along the A6 road, since this is the only existing connection to the FEZ territory.

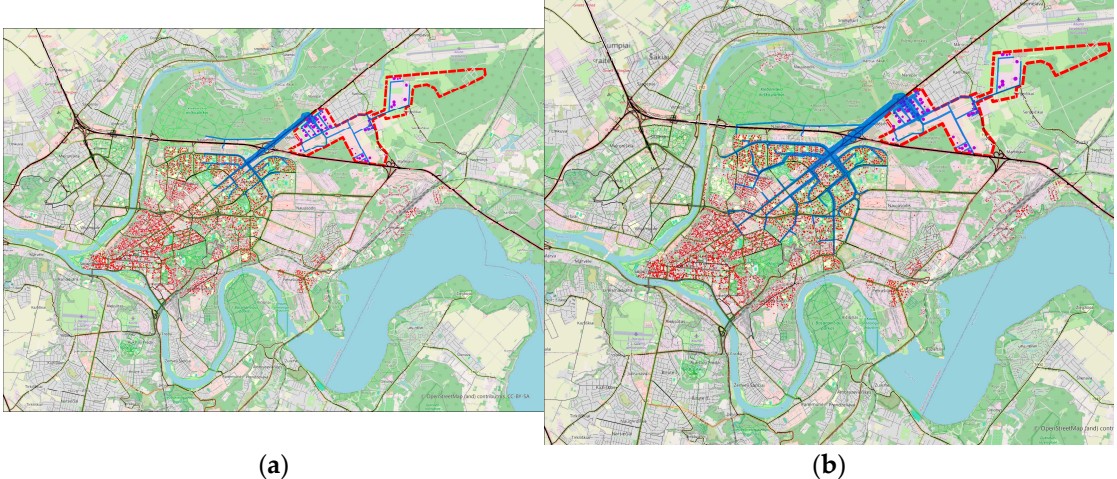

(**a**)                                                                                  (**b**)

**Figure 2.** *Status quo* situation: (**a**) generated bicycle flow within a 5 km distance; (**b**) generated bicycle flow within a 7 km distance (red dashed line marks the FEZ territory, blue lines represent traffic flow cartogram, red dots—locations of living places, and purple dots—locations of FEZ companies).

The initial decision matrix is outlined in Table 2, featuring four criteria expressed as grey values. The criterion "Population in reachable zone" encompasses a degree of uncertainty regarding the optimal distance for comfort, prompting calculations for both 5 km and 7 km distances to establish lower and upper values for the $X_1$ criterion. Consequently, this calculation yielded additional uncertain values for criteria $X_3$, $X_4$, and $X_{10}$.

Table 3 displays the normalized decision matrix, along with the corresponding weights of the criteria. Table 4 presents the normalized and weighted decision-making matrix, along with the rankings assigned to each alternative.

**Table 2.** Initial decision-making matrix.

| | $x_{1\alpha}$ | $x_{1\beta}$ | $x_2$ | $x_{3\alpha}$ | $x_{3\beta}$ | $x_{4\alpha}$ | $x_{4\beta}$ | $x_5$ | $x_6$ | $x_7$ | $x_8$ | $x_9$ | $x_{10\alpha}$ | $x_{10\beta}$ |
|---|---|---|---|---|---|---|---|---|---|---|---|---|---|---|
| | Population within 5–7 km distance, number of people | | Average Distance, m | Number of daily non-motorized commuters, number of people | | Reduction of traffic flow on A6 road, number of cars | | Savings from reduced GHG emissions, mln. €/year | Non-motorized travel distance, 1000 km/year | Savings from reduced traffic congestion, mln. €/year | Savings from reduced travel cost, mln. €/year | Cost, mln. € | Risk of Accident on dangerous A6 crossings, Number of possible non-car commuters | |
| $A_0$ | 23,825 | 85,882 | 5629 | 238 | 795 | 0 | 0 | 0 | 1342.5 | 0 | 0.1718 | 0 | 23,565 | 85,351 |
| $A_1$ | 44,331 | 109,438 | 5174 | 304 | 1013 | 66 | 218 | 0.3774 | 1572.4 | 0.7737 | 0.2013 | 1.263 | 20,077 | 64,129 |
| $A_2$ | 41,188 | 106,427 | 5232 | 295 | 985 | 57 | 190 | 0.3711 | 1546.1 | 0.6744 | 0.1979 | 1.266 | 11,180 | 34,666 |
| $A_3$ | 27,651 | 99,047 | 5602 | 275 | 916 | 37 | 121 | 0.3695 | 1539.4 | 0.4295 | 0.197 | 1.372 | 23,565 | 80,538 |
| $A_4$ | 24,656 | 95,301 | 5663 | 265 | 882 | 27 | 87 | 0.3596 | 1498.4 | 0.3088 | 0.1918 | 1.318 | 23,565 | 83,640 |
| $A_{12}$ | 45,435 | 110,329 | 5142 | 306 | 1021 | 68 | 226 | 0.378 | 1575 | 0.8021 | 0.2016 | 2.529 | 20,077 | 64,129 |
| $A_{13}$ | 47,011 | 114,200 | 5157 | 317 | 1057 | 79 | 262 | 0.3925 | 1635.3 | 0.9299 | 0.2093 | 2.635 | 20,077 | 64,129 |
| $A_{14}$ | 45,162 | 115,212 | 5207 | 320 | 1066 | 82 | 271 | 0.3996 | 1665.2 | 0.9618 | 0.2131 | 2.581 | 20,077 | 64,129 |
| $A_{23}$ | 43,202 | 110,738 | 5218 | 307 | 1025 | 69 | 230 | 0.3851 | 1604.5 | 0.8163 | 0.2054 | 2.638 | 11,180 | 34,666 |
| $A_{24}$ | 42,019 | 111,747 | 5261 | 310 | 1034 | 72 | 239 | 0.3917 | 1632 | 0.8483 | 0.2089 | 2.584 | 11,180 | 34,666 |
| $A_{34}$ | 28,249 | 102,548 | 5612 | 285 | 949 | 47 | 154 | 0.3834 | 1597.7 | 0.5466 | 0.2045 | 2.689 | 23,565 | 80,538 |
| $A_{134}$ | 47,609 | 117,696 | 5179 | 327 | 1089 | 89 | 294 | 0.4061 | 1692 | 1.0435 | 0.2166 | 3.953 | 20,077 | 64,129 |
| $A_{234}$ | 43,801 | 114,236 | 5239 | 317 | 1057 | 79 | 262 | 0.3987 | 1661.3 | 0.9299 | 0.2126 | 3.956 | 11,180 | 34,666 |
| $A_{1234}$ | 47,898 | 117,924 | 5154 | 327 | 1091 | 89 | 296 | 0.4049 | 1686.9 | 1.0506 | 0.2159 | 5.219 | 20,077 | 64,129 |

Note: Columns with greyed background represent Grey values.

**Table 3.** Normalized decision-making matrix.

| | $x_{1\alpha}$ | $x_{1\beta}$ | $x_2$ | $x_{3\alpha}$ | $x_{3\beta}$ | $x_{4\alpha}$ | $x_{4\beta}$ | $x_5$ | $x_6$ | $x_7$ | $x_8$ | $x_9$ | $x_{10\alpha}$ | $x_{10\beta}$ |
|---|---|---|---|---|---|---|---|---|---|---|---|---|---|---|
| Criteria weights | 0.16 | | 0.14 | 0.12 | | 0.12 | | 0.11 | 0.09 | 0.08 | 0.07 | 0.06 | 0.05 | |
| $A_0$ | 0.04316 | 0.05685 | 0.06741 | 0.01702 | 0.18960 | 0.00000 | 0.00000 | 0.06034 | 0.08427 | 0.00000 | 0.06033 | 0.0000 | 0.01418 | 0.16483 |
| $A_1$ | 0.08030 | 0.07244 | 0.07334 | 0.02175 | 0.24159 | 0.02316 | 0.25319 | 0.07068 | 0.07195 | 0.07649 | 0.07069 | 0.1305 | 0.01887 | 0.19346 |
| $A_2$ | 0.07461 | 0.07045 | 0.07252 | 0.02110 | 0.23492 | 0.02000 | 0.22067 | 0.06950 | 0.07317 | 0.06667 | 0.06949 | 0.1302 | 0.03491 | 0.34742 |
| $A_3$ | 0.05009 | 0.06556 | 0.06773 | 0.01967 | 0.21846 | 0.01298 | 0.14053 | 0.06920 | 0.07349 | 0.04246 | 0.06918 | 0.1201 | 0.01503 | 0.16483 |
| $A_4$ | 0.04466 | 0.06308 | 0.06700 | 0.01896 | 0.21035 | 0.00947 | 0.10105 | 0.06734 | 0.07550 | 0.03053 | 0.06735 | 0.1251 | 0.01447 | 0.16483 |
| $A_{12}$ | 0.08230 | 0.07303 | 0.07379 | 0.02189 | 0.24350 | 0.02386 | 0.26249 | 0.07079 | 0.07183 | 0.07929 | 0.07079 | 0.0652 | 0.01887 | 0.19346 |
| $A_{13}$ | 0.08516 | 0.07559 | 0.07358 | 0.02268 | 0.25209 | 0.02772 | 0.30430 | 0.07350 | 0.06918 | 0.09193 | 0.07350 | 0.0626 | 0.01887 | 0.19346 |
| $A_{14}$ | 0.08181 | 0.07626 | 0.07287 | 0.02289 | 0.25423 | 0.02877 | 0.31475 | 0.07483 | 0.06794 | 0.09508 | 0.07483 | 0.0639 | 0.01887 | 0.19346 |
| $A_{23}$ | 0.07826 | 0.07330 | 0.07272 | 0.02196 | 0.24446 | 0.02421 | 0.26713 | 0.07212 | 0.07051 | 0.08070 | 0.07213 | 0.0625 | 0.03491 | 0.34742 |
| $A_{24}$ | 0.07612 | 0.07397 | 0.07212 | 0.02217 | 0.24660 | 0.02526 | 0.27758 | 0.07335 | 0.06932 | 0.08386 | 0.07336 | 0.0638 | 0.03491 | 0.34742 |
| $A_{34}$ | 0.05117 | 0.06788 | 0.06761 | 0.02039 | 0.22633 | 0.01649 | 0.17886 | 0.07180 | 0.07081 | 0.05404 | 0.07181 | 0.0613 | 0.01503 | 0.16483 |
| $A_{134}$ | 0.08624 | 0.07791 | 0.07326 | 0.02339 | 0.25972 | 0.03123 | 0.34146 | 0.07605 | 0.06686 | 0.10316 | 0.07606 | 0.0417 | 0.01887 | 0.19346 |
| $A_{234}$ | 0.07934 | 0.07562 | 0.07243 | 0.02268 | 0.25209 | 0.02772 | 0.30430 | 0.07467 | 0.06810 | 0.09193 | 0.07466 | 0.0417 | 0.03491 | 0.34742 |
| $A_{1234}$ | 0.08677 | 0.07806 | 0.07362 | 0.02339 | 0.26020 | 0.03123 | 0.34379 | 0.07583 | 0.06707 | 0.10386 | 0.07582 | 0.0316 | 0.01887 | 0.19346 |

Note: Columns with greyed background represent Grey values.

Figure 2 shows the traffic flow graph in the status quo situation with 5 km and 7 km estimation distances. The only way from the living districts to FEZ is through the dangerous A6 road path. From these schemes, it also becomes clear that only one connection, shown in Figure 1 by number 1, is enough to achieve the best results. Other connections would work and be useful only after the future development and expansion of Kaunas FEZ.

Figure 3 shows the traffic flow graph with proposed new connections (all four connections, alternative A$_{1234}$) with 5 km and 7 km estimation distances. The diagrams demonstrate a more uniform distribution of traffic compared to the status quo scenario, along with a reduced reliance on the hazardous A6 path. These representations elucidate that the connection denoted as number 1 in Figure 1 experiences the highest usage, indicating that its construction would yield the most significant improvements in connectivity and safety. The efficacy of other connections is contingent upon the future development and expansion of Kaunas FEZ, particularly when companies occupy eastern territories.

**Table 4.** Normalized and weighted decision-making matrix.

| | | $x_{1\alpha}$ | $x_{1\beta}$ | $x_2$ | $x_{3\alpha}$ | $x_{3\beta}$ | $x_{4\alpha}$ | $x_{4\beta}$ | $x_5$ | $x_6$ | $x_7$ | $x_8$ | $x_9$ | $x_{10\alpha}$ | $x_{10\beta}$ | **Ki** | **Ui** | **Rank** |
|---|---|---|---|---|---|---|---|---|---|---|---|---|---|---|---|---|---|---|
| $A_0$ | 0.000000 | 0.006905 | 0.009096 | 0.009437 | 0.002043 | 0.022752 | 0.000000 | 0.000000 | 0.006637 | 0.007584 | 0 | 0.004223 | 0.000000 | 0.000709 | 0.008241 | 0.052755 | 0.535542 | **14** |
| $A_1$ | 0.007830 | 0.012849 | 0.011591 | 0.010267 | 0.002609 | 0.028991 | 0.002779 | 0.030383 | 0.007774 | 0.006475 | 0.006119 | 0.004948 | 0.007830 | 0.000943 | 0.009673 | 0.093324 | 0.947375 | **9** |
| $A_2$ | 0.007812 | 0.011938 | 0.011272 | 0.010153 | 0.002532 | 0.028190 | 0.002400 | 0.026481 | 0.007645 | 0.006586 | 0.005334 | 0.004865 | 0.007812 | 0.001745 | 0.017371 | 0.093358 | 0.947721 | **8** |
| $A_3$ | 0.007208 | 0.008014 | 0.010490 | 0.009483 | 0.002361 | 0.026215 | 0.001558 | 0.016864 | 0.007612 | 0.006614 | 0.003397 | 0.004843 | 0.007208 | 0.000751 | 0.008241 | 0.076403 | 0.775608 | **12** |
| $A_4$ | 0.007504 | 0.007146 | 0.010093 | 0.009380 | 0.002275 | 0.025242 | 0.001137 | 0.012125 | 0.007408 | 0.006795 | 0.002442 | 0.004715 | 0.007504 | 0.000723 | 0.008241 | 0.071735 | 0.728223 | **13** |
| $A_{12}$ | 0.003911 | 0.013169 | 0.011685 | 0.010331 | 0.002627 | 0.029220 | 0.002863 | 0.031498 | 0.007787 | 0.006465 | 0.006344 | 0.004956 | 0.003911 | 0.000943 | 0.009673 | 0.090631 | 0.920043 | **10** |
| $A_{13}$ | 0.003753 | 0.013625 | 0.012095 | 0.010301 | 0.002721 | 0.030250 | 0.003326 | 0.036516 | 0.008086 | 0.006226 | 0.007354 | 0.005145 | 0.003753 | 0.000943 | 0.009673 | 0.095440 | 0.968862 | **6** |
| $A_{14}$ | 0.003832 | 0.013090 | 0.012202 | 0.010202 | 0.002747 | 0.030508 | 0.003453 | 0.037770 | 0.008232 | 0.006115 | 0.007607 | 0.005238 | 0.003832 | 0.000943 | 0.009673 | 0.096418 | 0.978784 | **4** |
| $A_{23}$ | 0.003749 | 0.012521 | 0.011728 | 0.010180 | 0.002635 | 0.029335 | 0.002905 | 0.032056 | 0.007933 | 0.006346 | 0.006456 | 0.005049 | 0.003749 | 0.001745 | 0.017371 | 0.094862 | 0.962988 | **7** |
| $A_{24}$ | 0.003827 | 0.012179 | 0.011835 | 0.010097 | 0.002661 | 0.029592 | 0.003032 | 0.033310 | 0.008069 | 0.006239 | 0.006709 | 0.005135 | 0.003827 | 0.001745 | 0.017371 | 0.095939 | 0.973924 | **5** |
| $A_{34}$ | 0.003678 | 0.008188 | 0.010861 | 0.009466 | 0.002446 | 0.027160 | 0.001979 | 0.021463 | 0.007898 | 0.006373 | 0.004323 | 0.005027 | 0.003678 | 0.000751 | 0.008241 | 0.077309 | 0.7848 | **11** |
| $A_{134}$ | 0.002502 | 0.013799 | 0.012465 | 0.010257 | 0.002807 | 0.031166 | 0.003747 | 0.040976 | 0.008366 | 0.006018 | 0.008253 | 0.005324 | 0.002502 | 0.000943 | 0.009673 | 0.098508 | 1 | **1** |
| $A_{234}$ | 0.002500 | 0.012695 | 0.012099 | 0.010140 | 0.002721 | 0.030250 | 0.003326 | 0.036516 | 0.008213 | 0.006129 | 0.007354 | 0.005226 | 0.002500 | 0.001745 | 0.017371 | 0.097924 | 0.994073 | **3** |
| $A_{1234}$ | 0.001895 | 0.013883 | 0.012489 | 0.010307 | 0.002807 | 0.031223 | 0.003747 | 0.041254 | 0.008341 | 0.006036 | 0.008309 | 0.005307 | 0.001895 | 0.000943 | 0.009673 | 0.098205 | 0.996927 | **2** |

Note: Columns with greyed background represent Grey values. Bold column at the end of the table shows final result—the evaluated rank of alternative.

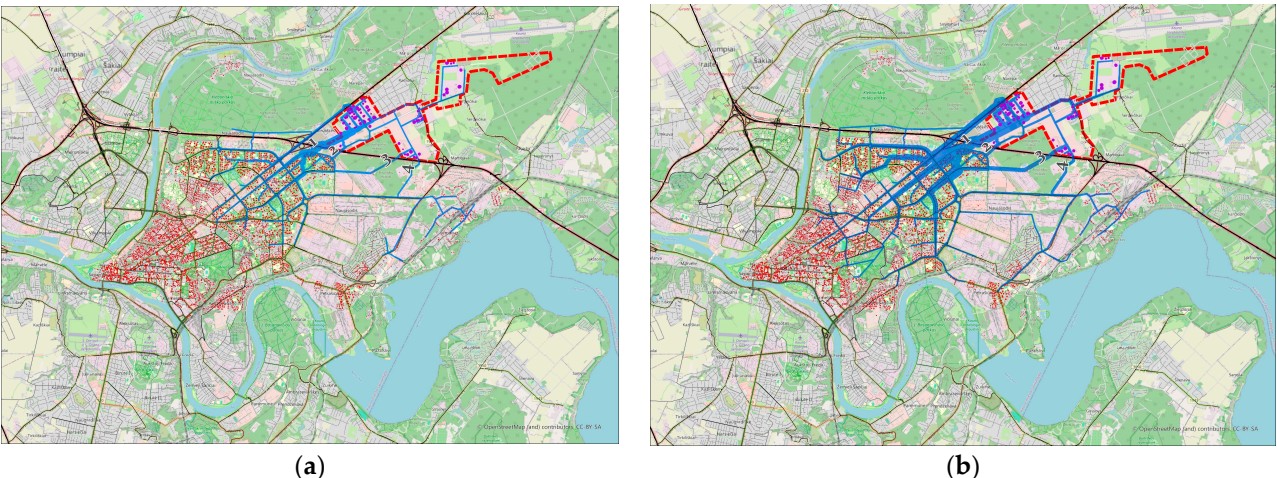

**Figure 3.** Proposed optimized bicycle pathway connections to the industrial zone: (**a**) generated bicycle flow within a 5 km distance; (**b**) generated bicycle flow within a 7 km distance (numbers represent the proposed highway crossing connections 1 to 4 as viaducts or tunnels, red dashed line marks the FEZ territory, blue lines represent traffic flow cartogram, red dots—locations of living places, and purple dots—locations of FEZ companies).

The findings indicate the potential to double the number of commuters upon establishing the missing links. For instance, when accounting for a closer commuting distance of 5 km, the actual number of commuters per day surges from 221 to 446. Conversely, if commuters can cover a larger distance of 7 km, the increase is less substantial—calculations reveal a rise from 797 to 1096, representing only a 37% increment. Nonetheless, this remains a considerable figure, achievable through the construction of two or even just one 50 m tunnel or viaduct crossing the highway at a strategic location.

Study results show that a significant increase in cycling and walking is possible if the connectivity of the cycling pathway network can increase significantly if proper connections crossing the highway would be established. Such investments have a high risk of not paying back; therefore, they are not welcomed by stakeholders and private companies.

## 7. Discussion

To stimulate investment, stimulate economic growth, and encourage job creation, many countries have established "zones of advantage" tailored for new business ventures. Currently, there are approximately 5400 industry zones worldwide [63]. offering various incentives such as tax breaks, financial assistance, and logistical support. However, the growth of these zones has paralleled an uptick in traffic volumes, leading to congestion and prolonged travel times.

The expansion of job opportunities in these zones necessitates improved accessibility. Implementing proper cycling and walking connections, alongside other infrastructure enhancements, is essential to reducing reliance on automobiles among workers. Emphasizing non-motorized travel modes aligns with the European Union's transport decarbonization objectives and discourages further expansion of car-centric areas.

The promotion of cycling and walking as preferred modes of transportation aligns closely with the principles of sustainable urban mobility, offering numerous benefits such as improved human health, energy conservation, and enhanced urban vibrancy. However, facilitating infrastructure for pedestrian and bicycle mobility poses a multifaceted challenge, particularly in industrial territories where highways often encircle these zones. Achieving comfortable and safe passage for cyclists and pedestrians across these highways is crucial to fostering increased cycling and walking activity.

In addition to the challenges posed by highway encirclement in industrial zones, there are several other factors that contribute to the complexity of providing pedestrian

and bicycle infrastructure in car-oriented industry zones. One such factor is the need to accommodate diverse user needs and preferences, including varying levels of cycling proficiency and mobility limitations. Ensuring that cycling and walking routes are accessible and user-friendly for individuals of all ages and abilities is essential to promoting their widespread adoption.

Furthermore, the integration of cycling and walking infrastructure must be accompanied by supportive policies and initiatives to encourage mode shift away from car dependency. This may include measures such as implementing traffic-calming measures, establishing pedestrian zones, and providing incentives for active transportation, such as bike-sharing programs or subsidies for purchasing bicycles.

The planning and design of cycling and walking infrastructure must consider broader urban planning objectives, such as promoting compact, mixed-use development and creating vibrant, livable communities. By incorporating cycling and walking infrastructure into larger urban design strategies, cities can enhance the overall quality of life for residents while reducing reliance on automobiles and mitigating the negative impacts of traffic congestion and air pollution.

Addressing concerns related to safety and security is paramount to encouraging greater uptake of cycling and walking. This involves not only designing infrastructure that minimizes conflicts between different modes of transportation but also implementing measures to improve lighting, signage, and visibility to enhance the perception of safety for cyclists and pedestrians.

Overall, while the task of enhancing pedestrian and bicycle connectivity in industrial zones may be complex, it presents an opportunity to advance broader goals of sustainable urban development, including promoting public health, reducing greenhouse gas emissions, and creating more resilient and equitable communities. By taking a holistic approach that considers the diverse needs of users and integrates cycling and walking infrastructure with supportive policies and urban design principles, cities can foster a more sustainable and livable built environment for all.

Safety concerns are paramount in urban planning, particularly in the integration of freight transport and bicycle lanes. This separation is a common practice in countries with a strong bicycle culture, such as the Netherlands, Denmark, and Belgium. Urban planners and policymakers can draw valuable lessons from successful examples in industrial zones like the eco-industrial park (EIP) in Kalundborg, Denmark, Kawasaki Eco-town in Japan [64], and the Antwerp seaport zones. These areas demonstrate the effective coexistence between bicycle lanes and freight transport, offering insights for improving safety and accessibility in similar urban environments.

This study acknowledges the varied locations of industrial areas, where many are positioned far from residential districts, posing challenges for integrating sustainable mobility solutions. Additionally, the configuration and dispersal of these industrial zones, along with physical barriers separating them from living areas, may present insurmountable obstacles. Furthermore, constraints such as restricted zone access and the impracticability of constructing segregated bicycle lanes in certain areas further complicate efforts to promote pedestrian and bicycle connectivity. These factors underscore the complexity of implementing sustainable transportation initiatives in car-oriented industrial parks.

It is imperative to underscore the necessity of augmenting the study's validity and reliability through the integration of comprehensive real-world data and the proposition of future research directions. Despite the valuable insights gained from the current investigation, there remains an exigency for further refinement and expansion of methodologies.

To fortify the study's conclusions, forthcoming research endeavors should prioritize the acquisition and assimilation of extensive empirical data reflective of existing traffic patterns, commuting behaviors, and infrastructure conditions within industrial zones. This empirical grounding will serve to refine models and simulations, leading to more precise assessments.

There is a need to explore the long-term implications and sustainability of proposed interventions. Through longitudinal analyses and scenario planning, researchers can

evaluate the enduring influence of implemented bicycle and pedestrian connections on travel behaviors, traffic flows, and environmental outcomes. Such an approach would offer invaluable insights into the efficacy and durability of proposed solutions, facilitating informed decision making by policymakers and urban planners.

Interdisciplinary collaboration and stakeholder engagement are pivotal for future research endeavors. By involving experts from diverse disciplines and engaging with local communities, businesses, and policymakers, researchers can gain multifaceted perspectives and address the complex challenges associated with sustainable mobility comprehensively. This collaborative approach will ensure that research findings are aligned with the needs and priorities of stakeholders, fostering greater acceptance and implementation of sustainable transportation initiatives.

## 8. Conclusions

Instances where job sites are isolated from residential areas are frequently encountered in industrial zones, often situated near highways in proximity to larger towns. The issue of bicycle and pedestrian accessibility to such zones has historically received limited attention from researchers or planners until recent years, coinciding with the emergence of sustainable traffic solutions. Even presently, established methods for assessing accessibility by bicycle or walking at the micro-scale remain elusive. This case study, employing a practical evaluation method, illustrates the opportunities afforded by innovations of the last decade, including enhanced computational capabilities and easily executable GIS data operations facilitated by web cloud computational platforms and advanced Python libraries.

The utilization of the Multi-Criteria Decision-Making (MCDM) ARAS-G method offers an objective and mathematically rigorous perspective on connection alternatives, allowing for weighted assessments that account for various criteria. Moreover, this method provides a structured approach to evaluating uncertainties, as it enables the incorporation of grey numbers for uncertain evaluation criteria. By leveraging these analytical tools, planners and decision makers can make informed choices regarding infrastructure development, fostering sustainable and efficient transportation solutions in industrial zones.

The study findings indicate the potential for a notable increase in cycling and walking if the connectivity of the cycling pathway network is substantially enhanced through the establishment of suitable connections crossing the highway. Furthermore, the Multi-Criteria Decision-Making (MCDM) analysis, incorporating expert rankings of criteria, underscores that investment in infrastructure to achieve sustainability and carbon-neutral transportation goals is primarily contingent on the usability of links rather than construction costs and economic benefits. Consequently, the best-ranked alternative is the one with connections 1, 3, and 4. Following closely is the alternative with all four connections. Interestingly, connections numbered 1 and 2, despite being positioned favorably, were ranked only 9 and 8 out of 14 alternatives, respectively, highlighting a nuanced evaluation process.

Study findings underscore the potential for a significant uptick in cycling and walking within Kaunas FEZ if the connectivity of the cycling pathway network is bolstered through the construction of additional links crossing the A1 highway. This enhancement could nearly double the opportunity for individuals to reside close to their workplaces, from 23,797 to 47,989. However, the direct return on investment in connecting bridges or tunnels may be limited. The benefits would be dispersed across society and the Kaunas FEZ community, necessitating government intervention rather than reliance on stakeholders or private companies.

While the current study lays a solid foundation for understanding the potential benefits of enhancing pedestrian and bicycle connectivity in car-oriented industrial parks, future research efforts must center on integrating more real-world data, conducting longitudinal analyses, and fostering interdisciplinary collaboration to advance knowledge and inform evidence-based decision making in urban mobility planning.

**Author Contributions:** J.Z.; Conceptualization, GIS-based methodology, Python scripting, software, validation and formal analysis, writing—original draft preparation, visualization, investigation,

resources, data curation. Z.T.; Conceptualization, MCDM methodology, validation and formal analysis, review and editing, supervision. All authors have read and agreed to the published version of the manuscript.

**Funding:** This research received no external funding.

**Data Availability Statement:** All the code with detailed step descriptions can be found within the Google Colab notebook here: https://colab.research.google.com/drive/1WrKfqFME3unLBnXtWWyo2 kw8nw22jzEb?usp=share_link (accessed on 12 March 2023).

**Acknowledgments:** Map data copyrighted OpenStreetMap contributors and available from https://www.openstreetmap.org (accessed on 12 March 2023).

**Conflicts of Interest:** The authors declare no conflicts of interest.

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
