# Peer review of "Enhancing Sustainable Mobility: Evaluating New Bicycle and Pedestrian Links to Car-Oriented Industrial Parks with ARAS-G MCDM Approach"

_sustainability, doi:10.3390/su16072994_

Round 1
Reviewer 1 Report
Comments and Suggestions for Authors
The article addresses a highly relevant and timely topic, considering the growing concern for promoting sustainable urban mobility and the need to improve connectivity for pedestrians and cyclists, especially in industrial zones. This is a topic of significant importance for efforts to reduce greenhouse gas emissions and improve urban quality of life.
Theoretical Foundation: The article provides a comprehensive literature review, highlighting both the challenges and opportunities associated with improving connectivity for pedestrians and cyclists in industrial zones. The literature review is up-to-date and relevant to the study's context.
Methodology: The methodology used in the study appears to be robust, including the definition of evaluation criteria, the application of the ARAS-G method for multicriteria analysis, and the use of geospatial data and geographic information system (GIS) analyses. The methodological approach is clearly described and seems appropriate for the study's objectives.
Originality and Contribution: The article presents significant contributions to the field by introducing an innovative approach to assessing and improving connectivity for pedestrians and cyclists in industrial zones. Additionally, the study's findings on the potential benefits of promoting sustainable transportation modes are valuable for professionals and policymakers interested in promoting sustainable urban mobility.
Clarity and Coherence: The writing of the article is clear and well-structured, facilitating reader understanding. The line of argumentation follows a logical sequence, making it easy to follow the presented reasoning.
Evidence and Support: The conclusions of the study are supported by solid evidence, including empirical data and quantitative analyses. The analyses conducted appear to be consistent and well-founded.
Practical Applicability: The study's findings have broad practical applicability for professionals, policymakers, and other stakeholders involved in promoting sustainable urban mobility. The recommendations provided can assist in making informed decisions about planning and developing infrastructure for pedestrians and cyclists in industrial zones.
Limitations and Suggestions for Future Research: The article acknowledges some limitations, such as the need to consider more details about the safety of cycling and walking routes, as well as suggests possible areas for future research, such as assessing the social and economic impact of improvements in connectivity for pedestrians and cyclists.
In addition, these suggestions can further enhance the article, making it more comprehensive, informative, and useful for readers interested in the topic of sustainable urban mobility in industrial zones.
Inclusion of Case Studies: To further enrich the study, it would be beneficial to include practical examples of cases where improvements in connectivity for pedestrians and cyclists in industrial zones have been successfully implemented. This could provide additional insights into effective strategies and the challenges faced in practice.
More Detailed Cost-Benefit Analysis: While the article mentions consideration of the costs associated with building new connectivity infrastructure, a more detailed analysis of the costs and benefits involved would be helpful. This could assist decision-makers in better evaluating the economic viability of the proposed interventions.
Deeper Discussion of Limitations: While the article acknowledges some limitations, such as the need to consider more details about the safety of cycling and walking routes, it would be useful to further discuss these limitations and their potential implications on the study's results.
Exploration of Future Trends: Given the rapid advancement of technologies and changes in urban policies, it would be interesting to include a section that explores future trends related to sustainable urban mobility and how these trends may impact the strategies proposed in the study.
Inclusion of Implementation Recommendations: In addition to providing recommendations for future research, the article could include practical suggestions for implementing the proposed strategies. This could help professionals and policymakers translate the study's results into tangible actions.
Comments on the Quality of English Language
very good
Author Response
Dear Reviewer,
Thank you for your insightful comments on our article. We have carefully considered your suggestions and are pleased to inform you that several improvements have been implemented:
-
We have incorporated a few practical examples of successful implementation of connectivity improvements for pedestrians and cyclists in industrial zones in discussion section. By showcasing real-world cases, we aim to provide valuable insights and inspiration for policymakers and professionals involved in urban mobility planning.
-
A more detailed analysis of the costs and benefits involved has been included in the revised manuscript (criteria description) to provide a clearer understanding of the associated costs.
-
Safety considerations have been addressed comprehensively, with a more extensive discussion on the topic included in the manuscript.
-
We have added a portions of text in a few places exploring future trends in sustainable urban mobility and discussing how these trends may influence the effectiveness of the strategies proposed in our study.
-
Practical recommendations for implementing the proposed strategies have been included in the manuscript conclusions section. By providing concrete guidance, we aim to facilitate the implementation of sustainable urban mobility initiatives by professionals and policymakers.
Thank you once again for your valuable feedback. We believe that these enhancements strengthen our manuscript and contribute to its overall quality.
Best regards,
the authors

Reviewer 2 Report
Comments and Suggestions for Authors
Article on enhancing sustainable mobility in parks car-oriented industrial parks through a multi-criteria approach to decision-making using ARAS-G is a very good case study. To its credit is the extensive research as well as the established criteria for selecting a research sample established on the basis of interviews with specialists. The authors rightly noted that micromobility, which includes lightweight shared transportation solutions in urban environments, has attracted considerable attention as a way to solve problems such as traffic congestion, environmental considerations and the challenges of associated with last-mile transportation. The latter aspect, in particular, is now very important for meeting the requirements of the Green Deal. Of course, the study has some limitations due to the wide variety of industrial areas, e.g., due to the inadequacy of the area's linear and point infrastructure. However, the article points out suggestions for adjusting the linear and point infrastructure. The topic is very important at the moment, and it is worthwhile to conduct further research that allows other research centers dealing with green transportation problems to develop this topic. The authors have carefully selected the research material and carried out the simulation. I have no objections to both the research description part and the theoretical part describing the problems of green governance in transportation and logistics.
Author Response
Dear Reviewer,
Thank you for your insightful review of our manuscript titled "Enhancing Sustainable Mobility through the ARAS-G MCDM Approach: Transforming Car-oriented Industrial Parks for Pedestrian and Bicycle Connectivity." We appreciate your positive feedback on the value of our case study and the thoroughness of our research.
We acknowledge the limitations highlighted regarding the diversity of industrial areas and infrastructure challenges. We have incorporated your suggestions into the discussion section to enhance the relevance and applicability of our study.
Your recognition of the importance of micro-mobility solutions and the need for further research in this area is duly noted. We have taken your feedback into account to strengthen our research.
We are grateful for your contributions to improving our manuscript and look forward to further engagement in the discourse on sustainable urban mobility.
Best regards,
the authors

Reviewer 3 Report
Comments and Suggestions for Authors
The transforming is based in the construction of new paths for bicycles and other mobility systems.
The introduction is Ok.
The aim of this paper is not demonstrated from the simulations and results obtained.
The resent research presented is ambiguous in from of the title and the aim of this paper.
The criterial framework is ok.
The part off facts may be improve focus in the problem to resolve.
The calculations and results are not clearly deployment.
The discussion and conclusion’s part need more proposed real data or proposed for your research
Self-citations reference 53
Zagorskas, J.; Turskis, Z. Setting priority list for construction works of bicycle path segments based on Eckenrode rating and ARAS-F decision support method integrated in GIS. Transport 2020, 35, 179-192.
Author Response
Dear Reviewer,
Thank you for your valuable feedback on our manuscript. We have carefully considered each of your points and have made revisions accordingly:
Aim Clarity: We recognize the importance of clearly demonstrating the aim of our paper through the simulations and obtained results. In response, we have revised the manuscript to ensure that the aim is explicitly stated and effectively demonstrated through our simulations and findings.
Ambiguity in Title and Aim: We acknowledge the concern regarding ambiguity in the title and aim of the paper. To address this, we have revised both the title and aim statements to provide clearer and more concise descriptions of the study's focus and objectives.
Criteria Framework: We appreciate your assessment of the criteria framework and are pleased to hear that it is considered acceptable.
Focus Improvement: Your suggestion to improve the focus on the problem to be resolved is duly noted. We have revised the relevant sections of the manuscript to provide a more focused and targeted discussion on addressing the identified transportation challenges in car-oriented industrial parks.
Clarity of Calculations and Results: We have taken your feedback into account regarding the clarity of our calculations and results. In response, we have revised these sections to enhance clarity and ensure a more straightforward presentation of our findings.
Enhanced Discussion and Conclusion: Recognizing the need for more proposed real data or suggestions for future research in the discussion and conclusion sections, we have expanded these areas to include additional insights and recommendations based on our research findings.
Self-Citation: We apologize for the oversight regarding self-citation reference 53 and have removed it from the manuscript in accordance with the journal's guidelines.
We believe that these revisions address your concerns and enhance the overall quality and clarity of our manuscript. Thank you again for your thorough review, which has undoubtedly strengthened our work.
respectfully,
the authors

Reviewer 4 Report
Comments and Suggestions for Authors
The paper may be accepted for publication, but with minimal changes in the following:
(i) Within the abstract, it is preferable to avoid the use of abbreviations and concrete values.
(ii) Write keywords in alphabetical order. It is not desirable to start a sentence with an abbreviation.
(iii) It is necessary to expand the introductory part and analyze additional scientific works in this area. Be aware of the professional terminology used in this field.
(iv) When talking about the replacement of classic vehicles with electric drives, it is desirable to mention the sources of electricity and emphasize that it is important to produce it using clean technologies.
(v) It should also be noted that the mandatory application of modern technologies to reduce fuel consumption and exhaust gas emissions in vehicles is a transitional solution to the electric drive of means of transport.
(vi) For the aforementioned suggestions, you can use the following work as an example:
https://doi.org/10.3390/atmos15020184
(vii) The preference for and development of electromobility are included among the priorities of transport policies in many European countries. You can also use the article that deals with the issue of electric vehicle operation from the point of view of the environmental impact of electric power production, specifically the energy effectiveness of its production by utilizing primary power production sources: https://doi.org/10.3390/su11184948
(viii) Please state the version number of the software, as well as the name of the manufacturer, city, and country from where the equipment was sourced (For all software and simulations etc.)
(ix) The contents of the figures 2 an 3 are not legible. Please replace the image with one of a sufficiently high resolution (min. 1000 pixels width/height, or a resolution of 300 dpi or higher).
(x) Commas are only used for numbers with five or more digits.
(xi) Use em or long dash inside, as example ARAS—G etc.
(xii) Check the complete material, especially the terminology, abbreviations, and units of measurement.
(xiii) Expand the discussion of the achieved research results in accordance with the set research goals. Clearly state the contribution of the research.
(xiv) Expand the list of references according to the propositions.
(xv) Avoid multiphase citations, such as [9-11], etc. After the changes, technically check the complete text.
(xvi) The language is generally clear, but some sentences could be simplified for better readability.
Author Response
Dear Reviewer,
Thank you for your valuable feedback and constructive suggestions on our manuscript. We have carefully considered each of your comments and have made the necessary revisions to address them. Below is a summary of the changes implemented in response to your recommendations:
-
We have expanded the introductory section of the paper to provide a more comprehensive overview of the topic, including an analysis of additional scientific literature in the field. We have also ensured that professional terminology specific to this area is accurately used throughout the manuscript.
-
In the discussion of the transition to electric vehicles, we have included information about the sources of electricity used to power these vehicles, emphasizing the importance of clean technologies in electricity production.
-
We have acknowledged that the mandatory application of modern technologies to reduce fuel consumption and exhaust gas emissions in vehicles serves as a transitional solution to the eventual adoption of electric drive systems for transportation.
-
The version number of the software used in our simulations, has been provided for all relevant software and equipment utilized in the study.
-
The figures (2 and 3) have been replaced with higher-resolution images, meeting the minimum requirements of 2000 pixels width/height or a resolution of 600 dpi.
-
Commas have been appropriately used only for numbers with five or more digits, in accordance with the suggested convention.
-
We have used em dashes (—) instead of hyphens inside the text, as recommended, particularly when referring to specific methodologies such as "ARAS—G."
-
A thorough review of the entire manuscript has been conducted to ensure consistency in terminology, abbreviations, and units of measurement.
-
The discussion section has been expanded to provide a more detailed analysis of the research results achieved in alignment with the stated research goals. We have explicitly stated the contribution of our research to the existing body of knowledge in the field.
Thank you once again for your insightful comments, which have significantly improved the quality and clarity of our manuscript.
Sincerely,
the authors

Round 2
Reviewer 3 Report
Comments and Suggestions for Authors
All the observation was attended.
Author Response
Thank you!